# Bio-Vitrimers for Sustainable Circular Bio-Economy

**DOI:** 10.3390/polym14204338

**Published:** 2022-10-15

**Authors:** Sravendra Rana, Manisha Solanki, Nanda Gopal Sahoo, Balaji Krishnakumar

**Affiliations:** 1School of Engineering, Energy Acres, University of Petroleum and Energy Studies (UPES), Bidholi, Dehradun 248007, India; 2School of Business, Energy Acres, University of Petroleum & Energy Studies (UPES), Bidholi, Dehradun 248007, India; 3Prof. Rajendra Singh Nanoscience and Nanotechnology Centre, Department of Chemistry, D.S.B. Campus, Kumaun University, Nainital 263001, India; 4College of Engineering, The Florida A&M University-Florida State University, 2525 Pottsdamer St., Tallahassee, FL 32310-6046, USA

**Keywords:** circular-economy, sustainable vitrimers, recycling, self-healing polymers

## Abstract

The aim to achieve sustainable development goals (SDG) and cut CO_2_-emission is forcing researchers to develop bio-based materials over conventional polymers. Since most of the established bio-based polymeric materials demonstrate prominent sustainability, however, performance, cost, and durability limit their utilization in real-time applications. Additionally, a sustainable circular bioeconomy (CE) ensures SDGs deliver material production, where it ceases the linear approach from production to waste. Simultaneously, sustainable circular bio-economy promoted materials should exhibit the prominent properties to involve and substitute conventional materials. These interceptions can be resolved through state-of-the-art bio-vitrimeric materials that display durability/mechanical properties such as thermosets and processability/malleability such as thermoplastics. This article emphasizes the current need for vitrimers based on bio-derived chemicals; as well as to summarize the developed bio-based vitrimers (including reprocessing, recycling and self-healing properties) and their requirements for a sustainable circular economy in future prospects.

## 1. Introduction

Thermoset polymers exhibit excellent thermal and mechanical properties; however, due to their permanent covalent crosslinked networks [1,2,3,4], thermoset polymers exhibit a lack of recycling and reusability behavior [5]. An enormous amount of synthetic polymer product has been used by modern society, most of which are derived from fossil fuels (also known as petro based polymers) [6]. This continuous requirement of synthetic polymers accelerated the fossil fuel exhaustion, and the disposal of these synthetic polymers causes severe damages to the environment [7]. Thus, biobased monomers have been deployed to control synthetic production. In addition, most available natural materials have been promising components to produce bio-based polymers in an efficient manner. However, the renewability of nature materials is still a concern for their utilization pervasively, where constant extraction was not able to restore the natural environment. Owing to this, severe concern is required to develop bio-based monomer-derived polymeric systems [8]. Significantly, the utilization of bio-based waste usage for polymer production would retain the environment effectively and reduce the waste processing problems. Globally, many government and non-government organizations are taking serious measures to reduce the use of synthetic polymers and devise natural feedstocks utilizing polymer materials with the advantages of reusability, reprocessability and recyclability [9,10]. The significance of polymeric materials can also be increased by their reusability and self-repairable properties, where the effective healing is helpful to regain the materials properties (mechanical, thermal, electrical) which are helpful in deferring the waste [11,12]. The self-healing of polymers can be achieved in two effective ways vis-à-vis intrinsic [13,14,15,16,17,18,19] and extrinsic [20,21,22,23,24] concepts, with intrinsic self-healing fascinating many investigators, [18,19,25] due to the absence of external agents (vascular and capsule-based networks). Intrinsic self-healing was attained via physical means [16,26,27,28,29,30,31]; hydrophobic, ionic, and hydrogen, etc.) and chemical [32,33,34] reversible reactions.

To promote the healing and mechanical properties, Wu et al. has developed polydimethylsiloxane–dithiothreitol block polymer chains, where ultrahigh mobility to promote healing speed and formed high density of hydroxyl and boronate ester dynamic cross-link ensures good mechanical strength (0.43 MPa) and 1500% stretchability. Along with fast healing (30 s after damage) at room temperature, the elastomer demonstrates excellent self-adhesiveness to various surfaces both in air and under water [35]. However, the conductivity of these conductive composite electrodes after healing is limited, and so therefore, the authors have further incorporated silver nanowires in a polyborosiloxane (PBS)/polydimethylsiloxane (PDMS) double-network (DN) matrix. The double network matrix is able to move not only silver nanowires on the composite surfaces, but also heavier conductive fillers such as silver microflakes embedded inside the bulk due to the highly viscous flow of PBS, yielding a bulk resistivity as low as 0.002 Ω cm and achieving 100% restoration of its original conductivity after damage without any external stimulus [36].

However, owing to higher bonding strength, chemical interactions are being investigated more often. Dissociative dynamic covalent adaptive network exchange (DDCAN) based Diels–Alder reaction is often studied, where developed thermosets exhibit self-healing functions; however, a loss in network integrity was observed [37,38,39]. To display structural integrity via a controlled density of cross-linking along with reprocessability and self-healing properties, associative dynamic covalent adaptive network exchanges (ADCAN) have recently been focused upon by researchers. The major difference and advantage of vitrimeric dynamic covalent adaptive networks have impeded network breakage due to their associative processing mechanism. At the same time, the disassociative process mechanism would relax the chain linkages during the presence of the stimulus and then rearrange their network structure. Thus, the ADCAN vitrimer system could have the maximum possibility of doing the exchanges without dimension failures [39,40,41,42,43].

In 2011, ADCAN exchange-based new polymers “vitrimers” were reported via transesterification exchange reaction [42]. This introduction by Leibler et al. has extended the thermal-based classification (thermoplastic and thermoset) in polymer material, where these materials exhibit durability/mechanical properties such as thermosets and processability/malleability such as thermoplastics [44,45,46]. In the case of vitrimers, the bonds get reform without failure, owing to their dynamic network, and the temperature at which chain exchange takes place, as well as crosses 10^12^ Pa.S viscosity denotes (T_v_) topology transition temperature [47,48] where a slow exchange reaction below T_v_ and a faster exchange reaction above T_v_ was observed. Therefore, T_v_ describes a lower and upper-temperature limit required for recycling and service of materials [49]. Depending on the position of T_v_ (after or before T_g_) tends to follow the Williams Landel Ferry (WLF) equation (before T_g_) and the Arrhenius equation (Equation (1)) (Figure 1) (after T_g_), where E_a_ denotes the minimum energy required to start the exchange reaction [50]. The covalent adaptive network exchange (ADCAN) behaviour of vitrimers allows for their reprocessing/recycling in most of the performed material [51,52,53].
τ* = τ_0_ exp (E_a_/RT)
where, E_a_ = activation energy; τ_0_ = characteristic relaxation time; τ* = relaxation time; T = temperature; R = gas constant.

An abundant valorization of biomass wastes (like food, agriculture, etc.) would be the prominent resource to perform or produce the bio-based polymer and the greater substitution for the conventional polymers. On account of the availability of monomers and their eco-friendly nature, bio-derived polymers have gained much attention in the past decade [54,55,56,57].

Furthermore, biopolymers have been developed using different bio-derivatives obtained through various pathways including resources such as plant oils, [58,59] lignite [60,61,62] saccharides, [63] and isosorbides [64,65]. Generally, biopolymers (or) bio-plastics are derived from different pathways like (a) extraction/ modification of bio-mass; (b) polymerization of bio-monomers; and (c) extraction from bio-organisms (Figure 2) [66]. For example, natural rubber is collected from “Hevea braziliensis” trees, where collected rubber requires processing with hardener to attain commercial products such as tires [67], whereas starch-derived polymers are prepared via a “gelatinization” process [68]. Moreover, the starch granules processed bio-polymers have been widely used in packaging industries, along with different plasticizers like glycerol, polyol and sorbitol [69]. Researchers have also focused on the development of bio-composites, including bio-derived or bio-blends consisting of different percentages of bio/synthetic derivatives [70,71].

However, the properties of bio-based monomer-derived polymer are still showing vulnerability in front of synthetic polymer. Owing to this, several research findings have been reckoned to achieve a better bio-based polymer. In that scenario, developed bio-based vitrimer material is providing an adequate commitment to reach the commercial polymer-like properties. Several reports have been discussed regarding the vitrimeric behavior in various polymers [50,72]; however, a limited discussion has been reported about the bio-derived chemicals/monomers based vitrimers, which may herald a sustainable circular bioeconomy. This review article covers bio-derived vitrimeric materials, including their source materials, their recycling and self-healing properties, including future perspectives aiming to attain a sustainable circular bioeconomy.

## 2. Sustainable Circular Bioeconomy

The traditional polymer circular economy (CE) continues to be challenging due to its reprocessing/recycle ability; also, at the same time, newly developed substitute materials have not expressed similar performance to conventional materials involved in contemporary applications. Hence, linear approaches such as “take-make-use-waste” have severely affected sustainability modules where non-renewable resources have been used at maximum levels [73]. In addition, sustainability is termed along with the circular economy paradigm in recent times, although material sustainability differs from CE material. The circular economy mainly focuses on the economic, environmental and social impacts, whereas sustainability is more about an ecological importance [74]. Globally, frameworks have been formed to enhance the sustainable environment. The United Nations (UN) has designed 17 sustainable development goals to be enforced in all countries in order to reach the goal of a sustainable society by 2030 [75]. The 12th goal therein meticulously described responsible production and consumption, which would be a stimulating factor for moving toward a sustainable circular economy [76]. Thus, to achieve a sustainable circular economy, the material should demonstrate ecologic inwardness for circular economy competence. Substantially, the circular economy provides sustainability as a small quotient; however, in the vast notion, major fossil resource utilization impacts the ecology. Thus, a sustainability circular bioeconomy has been recommended in recent times to perpetuate bio waste as a producible medium, where the waste burning could cause the severe CO_2_ emissions [77]. Owing to this, several bio polymer-related studies have been reported to be in-line with reuse, repair, recycle and reprocess ability. Significantly, 3R’s (Reduce-Reuse-Recycle) stated CE materials are inadequately included at real-time application than the reported materials, where it has occurred due to their lack of properties. Overall, a subtle approach is required to manufacture the sustainable circular bioeconomy to determine material for real-time application needs [78]. In that scenario, biovitrimer performance gives the strength to envision a sustainable circular bioeconomy. In the pinnacle, transcend that the close loop material has also extended its frontiers with an upcycle treatment for better results [79]. Overall, different bio-based vitrimer material details could diversify the field of vitrimer based sustainable circular bioeconomy; thus, this report describes the bio-vitrimers classification for the future embedment.

## 3. Classification of Bio-Vitrimers

Based on the involvement of renewable sources, vitrimeric materials can be classified as labelled in Figure 3 [55], where fully bio-based vitrimers are synthesized using only bio-derived chemicals, whereas in the case of partial bio-vitrimers, the network formation can be achieved using petroleum based molecules along with bio-derived chemicals.

### 3.1. Fully Bio-Based Vitrimer

#### 3.1.1. Lignin Derivatives

Lignin based materials have covered a large section of bio-vitrimeric materials, where in a very first report zinc catalyzed fully bio-based vitrimers were developed using epoxy monomers derived from sebacic acid, cured with lignin molecules decorated with ozone functional groups, and covalent network exchanges were achieved through transesterification reaction exchanges [80]. An increment of lignin content is useful to enrich hydroxyl/ester groups in crosslink network, helpful to improve the mechanical and thermal properties. The generated epoxy vitrimer exhibits a prevailed shape memory and self-healing behaviors at 190 °C (5 min; healing efficiency = 70%), and at 80 °C (recovery ratio = 87–97%), respectively, and hence demonstrated fast stress relaxation at above 160 °C through transesterification reaction. Furthermore, recovering adhesive behavior was investigated through lap shear test, where aluminum sheets were joined together. The epoxy vitrimer adhesive joint exhibited sufficient strength (6.5 MPa) comparable to commercially available EPO glues (8 MPa), where after breaking, the separated sheets were joined together at 190 °C. The repaired material (recovered via transesterification exchange) shows 5 MPa shear strength [80]. In addition, zinc acetylacetonate (Zn (acac)_2_) prompted poly (ethylene glycol) diglycidyl ether (PEG-epoxy) were treated with lignin derived polycarboxylic acid (L-COOH).

Rana et al. has reported a sustainable vitrimer, prepared by incorporating biomass-derived activated carbon (AC) filler into the epoxy matrix [81]. The epoxy vitrimer has been prepared in single step reaction with bisphenol A diglycidyl ether (BADGE) and 2- aminophenyl disulfide (2-AFD) at 80 °C for 15 min and then added a different percentage of activated carbon dispersed solution into that. Subsequently, the mixture was poured into the mould and cured at 150 °C for 5 h. After that, the disulfide exchanges promoted temperature-dependent self-healing observed at 80 °C for 5 min in pristine epoxy vitrimer, and the material had demonstrated a lower temperature self-healing at 70 °C for 5 min upon the addition of activated carbon. Healing efficiency evaluated via flexural studies highlighted a prominent recovery in vitrimer biocomposites with 1 wt% of AC, where 85% and 70% efficiency was exhibited after two consecutive healings (Figure 4). The obtained vitrimers were used as a coating material, displaying thermal self-healing via a transesterification process. The performed vitrimer was optimized with different percentages of PEG-epoxy/L-COOH (1:1, 1:1.5, 1:2) [82]. The prepared sample relaxation was analyzed through stress relaxation studies, where 1:1.5 composition exhibited slower relaxation than the 1:1 and 1:2 samples. Thus, higher hydroxyl and carboxyl content contained epoxy vitrimer 1:1 and 1:2 samples were helpful to achieve a self-healing efficiency 90% and 100%, respectively, at 200 °C for 30 min. At the same time, epoxy 1:1.5 had resulted 60% healing efficiency at same conditions, with an extrapolated activation energy (Ea) of the epoxy vitrimer 1:1 of 54.42 kJ/mol.

To demonstrate removability and repairability, the developed vitrimer polymer was coated onto tin plates, where the coating demonstrates almost 90% self-healing efficiency. To avoid the presence of toxic catalyst, a catalyst free mechanism has been developed [83]. In addition, lignin-based disulfide promoted catalyst free (vanillin-derived dialdehyde and amine monomers involved) bio-based vitrimer was demonstrated with prominent reprocessability and self-healing ability. The performed catalyst free vanillin vitrimer was optimized with different hardener molar ratios of tris (2-aminoethyl) amine and (4,4′-disulfanediyldianiline) [84].

#### 3.1.2. Fructose Derivatives

Fructose derived furan dialdehyde based vitrimers were developed when crosslinked with furan dialdehyde via imine exchange reactions, where vitrimeric materials demonstrate a fast stress relaxation (at r.t.) owing to imine reversible bonds (dynamic exchange). The activation energy (E_a_ = 64 KJ/mol) of the vitrimer network follows Arrhenius behaviour exhibited an energy lower than the reported polyimine dynamic [85,86,87,88,89,90,91] and vitrimer [41] networks activation energies, whereas T_g_ (−10 °C) of bio-based polyimine vitrimer was higher than the T_v_ (−60 °C). Reprocessing was demonstrated three times (at 120 °C for 10 min) without any significant loss of the mechanical properties [50]. Subsequently, sugar derived dimethyl-2,5-furan dicarboxylate (DMFD) include polyester-hydroxy urethanes was prepared without isocyanate. Transcarbomylation reactions based covalent dynamic exchange was observed for the developed non-isocyanate polyester-urethane (NIPHEU) vitrimer (Figure 5) [92].

The reported polymer exhibits good thermal stability including melting temperature (93 and 110 °C), as well as the onset degradation temperatures ranging from 170 to 220 °C, prompted by thermally-induced bond exchanges via transcarbamoylation mechanism.

#### 3.1.3. Soybean and Castor Oil

Owing to their abundant availability and better biocompatibility, plants oil derived polymers have often been discussed, with epoxidized soybean oil, a soybean oil derived commercially available at low cost. The design of a plant oil derived vitrimer with high glass transition temperature and mechanical strength represents a significant challenge. Liu and coworkers demonstrated a fully bio-based epoxy vitrimer, where the performed epoxy vitrimer was prepared from ESO and rosin derived Fumaropimaric acid (FPA) with zinc acetylacetonate catalyst, demonstrating exchange via transesterification reactions. The materials properties were optimized based on the involvement of FPA in network (ESO-FPA1.0, ESO-FPA0.8 and ESO-FPA0.6), with the ESO-FPA1.0 cured network exhibiting the higher Tg (65 °C). FPA derived ESO-FPA1.0 in vitrimer network attained a higher tensile strength (16.62 MPa) than the conventional citric acid cured ESO (0.6 MPa) networks. This might be due to the presence of a rigid hydrogenated phenanthrene ring and more reactive groups of FPA, resulted in higher crosslinking density. Shape memory behavior was described at 80 °C (which was above T_g_) for 30 min and deformation of the shape was analyzed at 160 °C for 30 min, resulting in the shape fixity ratio (R_f_) and shape recovery ratio (R_r_) of 98% and 89%, respectively. The performed vitrimer material was degraded with ethanol at 120 °C for 2 h, where ethanol hydroxyl groups were reacted with the ESO-FPA ester groups, and curing happened without the involvement of a catalyst. However, recycled vitrimer exhibits a reduction in T_g_ (65 °C to 30 °C) and it reproduces only 88% (16 MPa to 10.5 MPa) of its initial mechanical strength [93]. In addition, a catalyst free vegetable oil derived (epoxidized soybean oil (ESO) cured with 4,4′-dithiodiphenylamine (APD)) epoxy vitrimer network was demonstrated. Varying percentage of crosslinked networks were investigated including the different curing time at 180 °C. However, the ESOV-28 (28 h cured) specimen demonstrates prominent tensile strength owing to its high crosslink density. All the performed samples displayed a similar thermal stability (without constrain of gel fraction); hence, described catalyst-free bio vitrimer material promoted their exchanges via disulfide exchanges and reprocessed efficiently at 180 °C for 10 min under 20 MPa pressure [94].

### 3.2. Partially Bio-Based Vitrimers

#### 3.2.1. Lignin Based Derivatives

Lignin-derived eugenol-based epoxy vitrimers have been prepared by Zhang and coworkers, where transesterification reaction promoted covalent adaptive network exchanges were achieved. The synthesized eugenol epoxy was further reacted with succinic anhydride (SA) in the presence of zinc catalyst [95]. The performed eugenol derived epoxy vitrimer demonstrates self-healing at 190 °C, and shape memory and fast stress relaxation at 80 °C and 200 °C, respectively. Hence recycling of the material was achieved through physical and chemical principles, where the epoxy vitrimers were pocessed at 160 °C for 1h in hot press, and further the chemical degradation/decomposition of the polymer took place at 160 °C in the presence of ethanol (Figure 6). In another study, lignin derived vanillin based di-aldehyde monomer was treated with conventional diamine to obtain the corresponding bio-based vitrimers, where imine covalent exchange enables the physical reprocessing and chemical recycling. The dynamic imine metathesis reaction promotes self-healing at 180 °C for 1h. Cut sample shows optimal mechanical recovery after healing at 150 °C for 1 h (healing efficiency = 74.5%). At the same temperature, physical reprocessing was performed for 10 min, and the efficiency of reprocessed vitrimer was evaluated obtaining a 71.2% tensile strength and 72.8% elongation at break after three hot-pressing cycles. The chemical recycling was observed under an acidic environment at 50 °C, and the obtained aldehyde monomer can be reused again for the preparation of vitrimers [96].

#### 3.2.2. Isosorbide Derivative

Isosorbide derived monomers were reacted with aromatic diamines (4,4′-disulfanediyldianiline (MDS)) to obtain a ADCAN induced vitrimer material, where covalent network exchange takes place through a disulfide metathesis reaction. The resulting material demonstrate excellent thermomechanical properties in comparison to the conventional epoxy cured using 4,4′-methylenedianiline (MDA). The material exhibited a prominent reprocessing/self-healing at 100 °C for 60 min and shape memory at 80 °C for 1 min. The materials degrade in 5 wt% NaOH aqueous solution, owing to the presence of isosorbide [72].

## 4. Bio-Based Vitrimer Composites

The addition of carbon based nanofillers are helpful to obtain the thermal and photo induced transesterification based covalent exchanges in the bio-based epoxy networks. The addition of nanofillers is also helpful to rapid up the shape recovery, where NIR induced shape memory was observed even after fifth cycle (Recovery ratio = 100%), as well as a reduction in recovery time was observed due to light energy absorbed capacity of CNTs (which get convert into the thermal energy; E_a_ = 40.73 to 54.91 kJ/mol) [97]. The addition of nanofiller is not only helpful to improve the mechanical and thermal properties, it is also helpful to improve the adhesive fracture energy, as well as stress relaxation. The transcarbonation exchange based materials demonstrate prominent mechanical properties, as well as exhibiting reprocessing, self-healing and shape memory properties (Table 1), where tensile strength can be controlled based on the ratio between 1,3-Propanediol (PD) and bis (6-membered cyclic carbonate) (BCC). This change in tensile strength was observed due to their crosslink density, where PD soft segments were included in less; hence, the paper cellulose fiber network made the hydrogen bond interaction with covalently crosslinked vitrimer network helpful to enhance the mechanical properties. The reported material exhibits self-healing and shape memory at 150 °C, whereas, separately cut samples were reconnected together at 170 °C at 4 MPa, resulted a 80% healing efficiency [98].

## 5. Research Gap and Limitations

Prudently, bio-vitrimer is a promising material to achieve the sustainable circular bio-economy; however, more investigation is required to include these materials in real-time applications. Constantly, several polymer materials have been experimented with vitrimer mechanisms, though the performed material excessively carried out their exchanges via trans esterification reaction; specifically, some of them required the catalyst to exhibit the covalent exchanges. Owing to this, more relevant vitrimer mechanism executed material should be performed with a bio-based monomer. A material with extensive properties is always required to epitome for future perpetuation. On account of this, bio-based vitrimer material would have been only prepared with the bio feedstocks, and their wastes could not be part of an efficient practice to achieve a sustainable circular bio-economy. Along with that, catalyst free self-healing, recycling and reusability of the vitrimer mechanism-included upscaling is highly demandable. Altogether, for the upcoming requirements of the material world, these bio-based vitrimer materials could be developed with the catalyst-free exchange mechanism with low-temperature self-healing. Also, nanofiller addition-based studies would enhance their inclusion in a wide range of applications.

## 6. Outlook and Prospective

Renewable resource based materials are gaining much attention in an attempt to achieve sustainability objectives [54]. These bio-based recyclable polymeric materials are highly recommendable as a solution for plastic pollution. Instead of the conventional (thermoset and thermoplastic) synthetic polymers, bio-based vitrimers could be a future generation sustainable polymer, helpful in resolving the disposal challenges of polymeric materials. However, for practical applications, the poor mechanical and thermal stability of bio-polymers is prejudicial. Therefore, it is desirable to develop a catalyst free controlled cross-linked density bio-polymers including stress-relaxation ability, softening temperature, and activation energy for bond exchange with good thermal and mechanical properties. Given the large amount of thermoset-materials in polymer technology, the perspective to use recyclable materials in the vitrimeric concept is attractive, as modern regulations for polymers, their recyclabilities, together with the need to reduce CO_2_-emission, is pressing. Catalyst-free vitrimers can provide a temperature dependent recycling of many of the thermosetting polymers currently in use. Specifically, for accomplishing sustainability development goals and also predominantly for the sustainable circular bio economy, more efforts are required for the development of new smart materials with 3R’s (Reduce-Reuse-Recycle) properties. Society has opined that self-enrichment is inadequate to make for a better life; consequently, the concern about sustainability leads the globe toward a grave situation. Measures, however, have been taken to overcome this, as well as to ensure a flourishing future environment. Altogether, bio vitrimer systems are a promising material for achieving sustainable circular bioeconomy goals.

## 7. Conclusions

Comprehensively, this article covers bio-based vitrimer and provides research insights into the sustainable circular economy and future needs. To be precise, bio-based derived material and their incorporation with a vitrimer mechanism have been detailed here with evident outcome properties in a certain condition. Additionally, bio-based vitrimeric composite exhibited enhancements in properties and thus could be further explored. Adequate research is required in this area to meet and establish a sustainable circular bioeconomy.

## Figures and Tables

**Figure 1 polymers-14-04338-f001:**
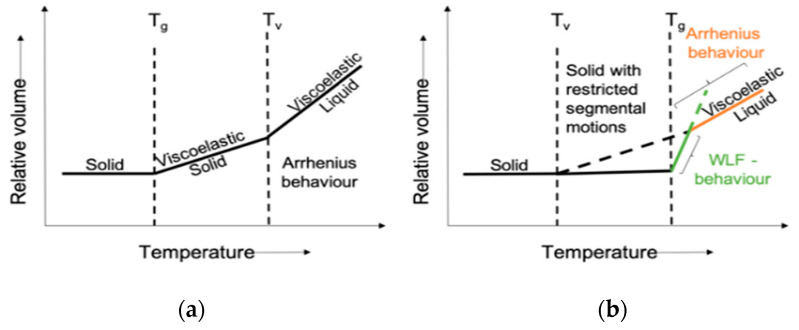
Temperature dependent phase transitions of vitrimers. (**a**) T_g_ below T_v_ and (**b**) T_v_ above T_g_. Reproduced with permission [52].

**Figure 2 polymers-14-04338-f002:**
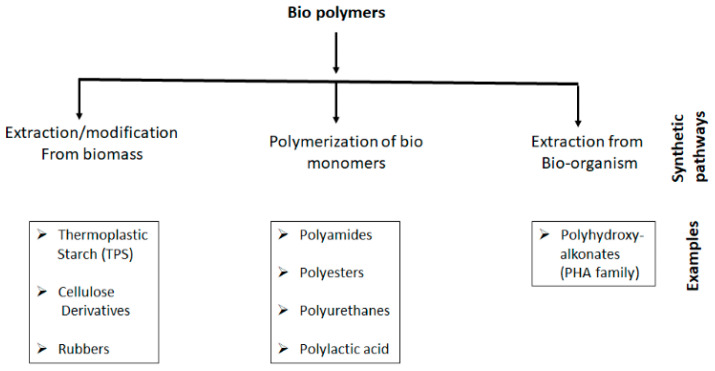
Biopolymers derived from different synthetic pathways.

**Figure 3 polymers-14-04338-f003:**
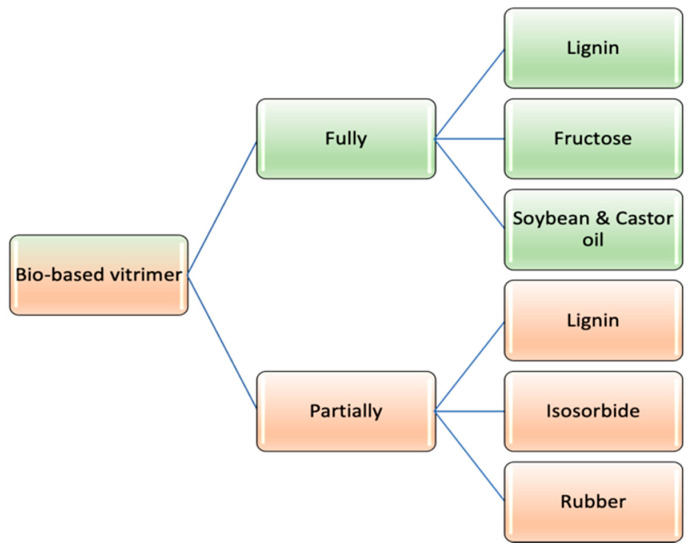
Bio-based vitrimer categorization.

**Figure 4 polymers-14-04338-f004:**
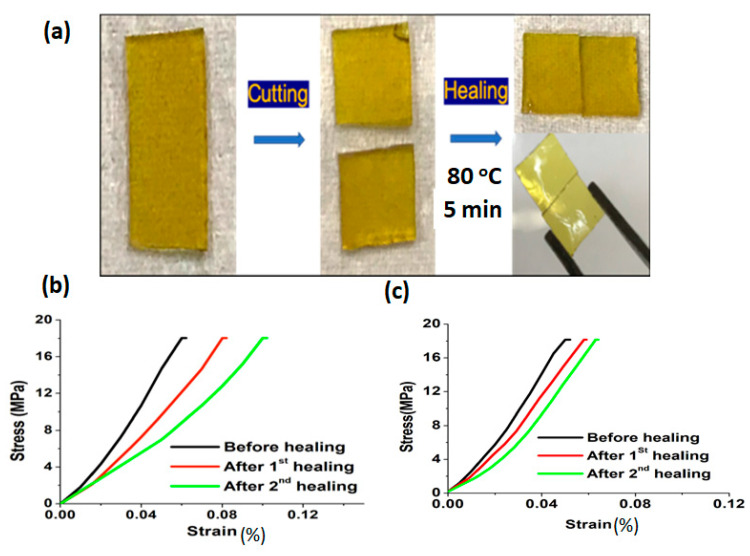
(**a**) Self-healing of the epoxy vitrimer (i) pristine EP-p, (ii) cut into two pieces, and (iii) rejoined. Healing efficiency of vitrimer was calculated via the stress-strain relationship for (**b**) EP-p and (**c**) EP-1. Reproduced with permission [81].

**Figure 5 polymers-14-04338-f005:**
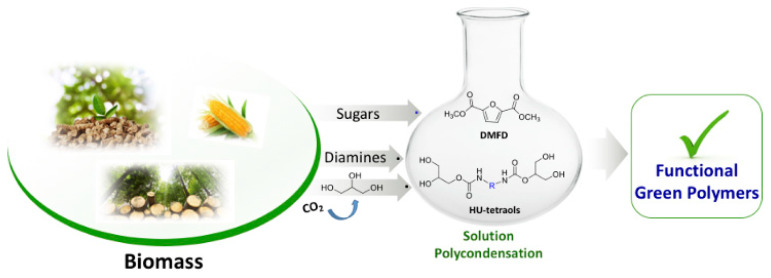
A green strategy for the synthesis of nonisocyanate polyester-urethanes. Reproduced with permission [92].

**Figure 6 polymers-14-04338-f006:**
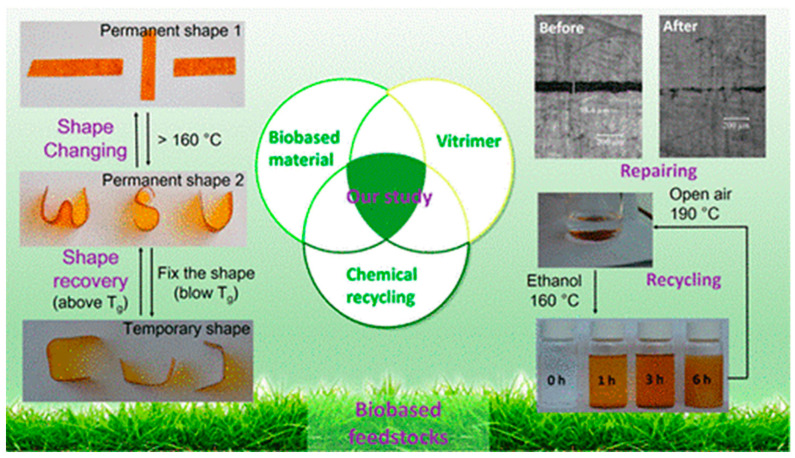
Eugenol-derived biobased epoxy. Reproduced with permission [95].

**Table 1 polymers-14-04338-t001:** Different bio-based vitrimers.

Bio-Based Derivatives	Material	Recycling/ Reprocessing	Self-Healing	Shape Memory	Ref.
Lignin based	Epoxy derived from eugenol with succinic anhydride	190 °C for 1 h, Low efficiency ^a,#^	190 °C for 1 h, 90% ^b^	80 °C for less than minute, 100% ^c^	[95]
	Dialdehyde monomer derived from vanillin with conventional diamine	150 °C for 10 min, 71.2% ^a,^*	150 °C for 1 h, 74.5% ^b^	-	[96]
	Dialdehyde derived from vanillin and amine monomers	60 °C for 20 min, Maximum efficiency ^a,#^	Addition of ethylene diamine	-	[84]
	Epoxy derived from sebacic acid and ozone crafted lignin	-	190 °C for 5 min, 70% ^b^	80 °C for less than minute, 87–97% ^c^	[80]
Fructose	Furan dialdehyde and fatty acid-based diamine/ triamine	120 °C for 10 min, Nearly 100% ^a,^*	-	-	[50]
Soybean & Castor oil	Fumaropim-aric acid (FPA) derived from Rosin and epoxidized soybean oil (ESO)	120 °C for 2 h, 88% ^a,^*	180 °C for 60 min, Nearly 100%	80 °C for 30 min, 89%	[93]
	4,4′-dithiodiphenylamine (APD) cured Epoxidized soybean oil (ESO)	180 °C for 10 min under 20 MPa, 80% ^a,~^	-	-	[94]
	Vinylogus urethane vitrimer derived from aminate DL-limonene (AL)	160 °C, 6 MPa for 30 min	-	70 °C for 1 min, 100%	[99]
Isosorbide	Isosorbide derived epoxy and aromatic diamines	100 °C for 1 h, 82.6% ^a,^*	100 °C for 1 h, 100% ^b^	80 °C for 1 min 100%	[72]
Natural rubber	Dodecanedioic acids (DAs) and aniline trimer (ACAT) derived epoxidized natural rubber	200 °C for 20 min, 88% ^a,#^	NIR and 200 °C for 30 min, 80% ^b^	NIR and 80 °C for less than minute, 95% ^c^	[100]
Composite	Cellulose paper from 1,3-Propanediol (PD) and bis (6-membered cyclic carbonate) (BCC) and	HCl at 90 °C for 36 h	160 °C for 10 s, 75% ^b,^*^,^^ and 170 °C for 2 h, 4 MPa, 80% ^b,^*^,$^	150 °C for less than minute ^c^	[98]
	Carbon nano tubes (MWNT) with epoxy/cashew nutshell liquid	-	-	NIR and 60 °C for less than minute, 100% ^c^	[97]

[^a^]-recovery efficiency. [^b^]-healing efficiency. [^c^]-shape recovery ratio. [*]-tensile strength. [^#^]-stress-strain. [^^^]-scratch. [^$^]-cut and overlapped. [^~^]-elongation at break.

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
