# Peer review of "Bio-Vitrimers for Sustainable Circular Bio-Economy"

_polymers, 2022, doi:10.3390/polym14204338_

Round 1

Reviewer 1 Report

I feel that the manuscript has grammatical, syntax, and spelling issues that need to be carefully addressed. Sentence structure and flow are also often problematic.

1)      Abstract: line 13; as well as to

2)      Abstract; lines 16-17; a sustainable circular

3)      Abstract: line 19; substitute the place

4)      Abstract: line 22; need for

5)      Keywords should be revised. The title words should not be repeated here.

6)      The terminology section is missing.

7)      What are the major findings and how they are addressing the left-behind research gaps and current challenges? This section should be highlighted.

8)      What needs to be done in future research? This question should be answered in this section. Also, the limitations of the current research are not clear.

9)  Abbreviations must be spelled out the first time they are mentioned in the abstract and starting again with the introduction section which includes elements, chemical names, etc.

Author Response

Comments and Suggestions for Authors

I feel that the manuscript has grammatical, syntax, and spelling issues that need to be carefully addressed. Sentence structure and flow are also often problematic.

1)      Abstract: line 13; as well as to

2)      Abstract; lines 16-17; a sustainable circular

3)      Abstract: line 19; substitute the place

4)      Abstract: line 22; need for

Response: As per the reviewer's suggestions, the grammatical, syntax and spelling issues have been checked and corrected in the revised draft.

5)      Keywords should be revised. The title words should not be repeated here.

Response: Thank you for the valuable comment. The new keywords have been incorporated into the revised draft.

6)      The terminology section is missing.

Response: Terminology section is added in the revised manuscript (Page 14).

7)      What are the major findings and how they are addressing the left-behind research gaps and current challenges? This section should be highlighted.

Response: As per the reviewer's suggestion, a new section (research and current challenges) has been included in the revised draft (Page No- 13; Line No-413-429).

Prudently, bio-vitrimer is a promising material to achieve the sustainable circular bio-economy, erstwhile before that, some more investigation is required to include these materi-als in real-time applications. Constantly, several polymer materials experimented with vitrimer mechanism, though the performed material excessively carried out their exchanges via transesterification reaction; especially, some of them needed the catalyst to exhibit the covalent ex-changes. Owing to this, a few more relevant vitrimer mechanism executed material should be performed with bio-based monomer. Infer that, a material with extensive properties is always required to epitome for future perpetuation. On account of this, bio-based vitrimer material would have been only prepared with the bio feedstocks and their wastes could not be the efficient practice to achieve the sustainable circular bio-economy. Thus, it should evidently be shown the more prominent properties than the contemporary materials, then the future needs would emphasise the bio-based vitrimer inclusion in the real-time application. Along with that, catalyst free self-healing, recycling, and reusability of the vitrimer mechanism included upcy-cling is highly demandable. Altogether, for the upcoming requirements of the material world, these bio-based vitrimer materials could be developed with the catalyst-free exchange mecha-nism with low-temperature self-healing would effectuate. Also, nanofiller addition-based stud-ies would enhance their inclusion in wide range of applications.

8)      What needs to be done in future research? This question should be answered in this section. Also, the limitations of the current research are not clear.

Response: As covered in comment 6, future research has been discussed in the revised draft (Page No- 13; Line No-413-429).

9)  Abbreviations must be spelled out the first time they are mentioned in the abstract and starting again with the introduction section which includes elements, chemical names, etc.

Response: As per the reviewer's recommendation, the abbreviations have been detailed in the body of the manuscript.

Reviewer 2 Report

Explain what do you mean by fully or partially in Fig.3

Author Response

Reviewer #2:

  1. Explain what do you mean by fully or partially in Fig.3

Response: Thank you for your valuable comment. A more detailed explanation has been incorporated in the revised manuscript (Page 5; line 198-202).

Based on the involvement of renewable sources, vitrimeric materials can be classified as labelled in Figure 3, [55] where fully bio-based vitrimers are synthesized using only bio-derived chemicals, whereas, in case of partial bio-vitrimers, the network formation can be achieved using petroleum based molecules along with bio-derived chemicals.

Reviewer 3 Report

With the arising concerns of environment, people care more about renewable materials. Bio-based materials as alternative has been widely applied to solve the polymeric waste caused by conventional materials. Bio-vitrimeric materials based on dynamic covalent adaptive network exchanges not only substitute the materials from original source but also endow the materials self-healing properties and improve their durability. The authors review the recent progress of bio-vitrimeric materials with different synthetic pathways. They first introduce the principle of preparation of bio-vitrimeric materials. Then bio-based vitrimer categorization from lignin, fructose, and soybean oils, as well as partially bio-based vitrimers were systematically investigated. Subsequently, sustainable circular bioeconomy was highlighted in the manuscript. Finally, the authors outlined the bio-based recyclable polymeric materials. This manuscript needs some revisions, and some suggestions were listed below for the improvement of this manuscript

1. Different types of bio-vitrimeric materials were introduced in the manuscript, however only transesterification exchange reaction was emphasized, other types of dynamic bonds should be introduced in the review.

2. What’s difference of the renewable sources and some information about these natural materials should be added.

3. Sustainable circular bioeconomy focus on the importance of sustainable materials, the reviewer thinks this paragraph is better to move to the front part before “Classification of bio-vitrimers” and how to connect these two parts should be considered?

4. What’s the difference between dynamic covalent adaptive network exchanges (ADCAN) and vitrimer?

5. Why introducing the addition of filler in the outlook part, it is better to introduce separately?

6. The present challenges and corresponding solution are not presented in the outlook, it is better to provide.

7. The quality of the figures is needed to improve.

8. In page 2, this sentence “An abundant valorization of……[54]” is repeated.

9. The experimental process to prepare polymer should not be introduced in reviewed (page 6).

10. Some important papers about self-healing materials are missing (Journal of Materials Chemistry A, 2019, 7(48): 27278-27288 and Journal of Materials Chemistry A, 2022, 10(4): 1750-1759).

Author Response

Reviewer #3:

  1. Different types of bio-vitrimeric materials were introduced in the manuscript, however only transesterification exchange reaction was emphasized, other types of dynamic bonds should be introduced in the review.

Response: Thank you for the valuable recommendation. However, till reported most of the vitrimer systems were performed with the transesterification exchange mechanism. Owing to that, this article emphasizes the same, although this manuscript has covered some other exchange mechanisms like disulfide, tanscarbomylation and transcarbonation, utilized for the development of bio-vitrimers.

  1. What’s difference of the renewable sources and some information about these natural materials should be added.

Response: As per the reviewer’s suggestion, some more information has been added for the same in the revised draft (Page No- 1; Line No- 37-45).

Thus, biobased monomers have been deployed to control synthetic production. In addition, most available natural materials have been promising components to produce bio-based polymer in an efficient manner. However, the renewability of nature materials is still a concern for the utilization of them pervasively, where constant extraction could not able to restore the nature environment. Owing to this, severe concern is required to develop bio-based monomer-derived polymer systems.[8] Significantly, the utilization of bio-based waste usage for polymer production would retain the environment effectively and reduce the waste processing problems.

  1. Sustainable circular bioeconomy focus on the importance of sustainable materials, the reviewer thinks this paragraph is better to move to the front part before “Classification of bio-vitrimers” and how to connect these two parts should be considered?

Response: We highly appreciate reviewer’s suggestion, thus, we have rearranged the section and connected two parts eloquently in the revised draft (Page No- 5; Line No- 164-193).

  1. What’s the difference between dynamic covalent adaptive network exchanges (ADCAN) and vitrimer?

Response: Thank you for the valuable recommendation. Difference between them have been detailed in the revised introduction draft (Page No- 2; Line No- 74-86).

Owing to higher bonding strength chemical interactions are investigated more often. Dissociative dynamic covalent adaptive network exchange (DDCAN) based Diels-Alder reaction is often studied, where developed thermosets exhibit self-healing functions, however, a loss in network integrity was observed.[37][38][39] To display structural integrity via a controlled density of cross-linking along with reprocessability and self-healing properties, associative dynamic covalent adaptive network exchanges (ADCAN) are recently focused by the researchers. The major difference and advantage of vitrimeric dynamic covalent adaptive networks have impeded network breakage due to their associative processing mechanism. At the same time, the dissociative process mechanism would relax the chain linkages, during the presence of the stimulus and then rearrange their network structure. Thus, ADCAN vitrimer system could have the maximum possibility to do the exchanges without dimension failures.

  1. Why introducing the addition of filler in the outlook part, it is better to introduce separately?

Response: Thank you for the valuable recommendation. Separate section has been included in the revised draft (Page No- 11; Line No- 366-382).

The addition of carbon based nanofillers are helpful to obtain the thermal and photo induced transesterification based covalent exchanges in the bio-based epoxy networks. The addition of nanofillers is also helpful to rapid up the shape recovery, where NIR induced shape memory was observed even after fifth cycle (Recovery ratio= 100%) as well as a reduction in recovery time was observed due to light energy absorbed capacity of CNTs (which get convert into the thermal energy; Ea= 40.73 to 54.91 kJ/mol).[98] The addition of nanofiller is not only helpful to improve the mechanical and thermal properties, it is also helpful to improve the adhesive fracture energy as well as stress relaxation. The transcarbonation exchange-based materials demonstrate prominent mechanical properties, as well as exhibit reprocessing, self-healing and shape memory properties (Table 1), where tensile strength can be controlled based on the ratio between 1,3-Propanediol (PD) and bis (6-membered cyclic carbonate) (BCC). This change in tensile strength was observed due to their crosslink density, where PD soft segments were included in less, hence paper cellulose fiber network made hydrogen bond interaction with covalently crosslinked vitrimer network, helpful to enhance the mechanical properties. The reported material exhibits self-healing and shape memory at 150 0C, whereas, separately cut samples were reconnected together at 170 ° at 4 MPa, resulted a 80% healing efficiency.[97]

  1. The present challenges and corresponding solution are not presented in the outlook, it is better to provide.

Response: As per the reviewer’s suggestion, research gap, limitation and future work has been discussed in the revised draft (Page 13, line 413-429).

Prudently, bio-vitrimer is a promising material to achieve the sustainable circular bio-economy, erstwhile before that, some more investigation is required to include these materials in real-time applications. Constantly, several polymer materials experimented with vitrimer mechanism, though the performed material excessively carried out their exchanges via transesterification reaction; especially, some of them needed the catalyst to exhibit the covalent ex-changes. Owing to this, a few more relevant vitirimer mechanism executed material should be performed with bio-based monomer. Infer that, a material with extensive properties is always required to epitome for future perpetuation. On account of this, bio-based vitrimer material would have been only prepared with the bio feedstocks and their wastes could not be the efficient practice to achieve the sustainable circular bio-economy. Thus, it should evidently be shown the more prominent properties than the contemporary materials, then the future needs would emphasise the bio-based vitrimer inclusion in the real-time application. Along with that, catalyst free self-healing, recycling and reusability of the vitrimer mechanism included upscaling is highly demandable. Altogether, for the upcoming requirements of the material world, these bio-based vitrimer materials could be developed with the catalyst-free exchange mechanism with low-temperature self-healing would effectuate. Also, nanofiller addition-based studies would enhance their inclusion in wide range of applications.

  1. The quality of the figures is needed to improve.

Response: Thank you for the suggestion. The figure qualities have been corrected in the revised draft.

  1. In page 2, this sentence “An abundant valorization of……[54]” is repeated.

Response: Thank you for the notification. Repeating sentence has removed from the revised draft.

  1. The experimental process to prepare polymer should be introduced in reviewed (page 6).

Response: Thank you for the valuable comment. Polymer preparation process has been incorporated in the revised draft (Page No- 7; Line No- 256-260).

The epoxy vitrimer has been prepared in single step reaction with bisphenol A diglycidyl ether (BADGE) and 2- aminophenyl disulfide (2-AFD) at 80 0C for 15 mins and then added a different percentage of activated carbon dispersed solution into that. Subsequently, the mixture was poured into the mould and cured at 150 0C for 5h.

  1. Some important papers about self-healing materials are missing (Journal of Materials Chemistry A, 2019, 7(48): 27278-27288 and Journal of Materials Chemistry A, 2022, 10(4): 1750-1759).

Response: Thank you for the valuable recommendation. The mentioned articles have been discussed effectively in the revised draft (Page No- 2; Line No- 60-73).

Reviewer 4 Report

Reorganization of the introduction to include a brief comparison of Vitrimers types, including bio-Vitrimers.

Replace figures with higher resolution ones.

Even if the article is a review, it needs conclusions.

Author Response

Reviewer #4:

  1. Reorganization of the introduction to include a brief comparison of Vitrimers types, including bio-Vitrimers.

Response: Thank you for the valuable inputs. The suggested changes were carried out in the Introduction section.

  1. Replace figures with higher resolution ones.

Response: Thank you for the suggestion. The figure qualities have been corrected in the revised draft.

  1. Even if the article is a review, it needs conclusions.

Response: As per the reviewer’s suggestion, conclusion section has been discussed in the revised draft (Page No- 14).

Round 2

Reviewer 1 Report

This paper can be accepted after minor editing of the English language.

Reviewer 4 Report

Suggestions have been included in the revised version; the manuscript can now be accepted for publication.

Best Regards for authors